# Bioelectric Effect of a Microcurrent Toothbrush on Plaque Removal

**DOI:** 10.3390/ijerph18168255

**Published:** 2021-08-04

**Authors:** Ji-Hyun Lee, Jin-Hee Ha

**Affiliations:** 1Department of Periodontology, Ulsan University Hospital, College of Medicine, University of Ulsan, Ulsan 44033, Korea; 2Department of Dentistry, Ulsan University Hospital, College of Medicine, University of Ulsan, Ulsan 44033, Korea; 0732289@uuh.ulsan.kr

**Keywords:** biofilm, bioelectric effect, plaque control, microcurrent therapy

## Abstract

This study evaluated the effectiveness of a microcurrent toothbrush (approved by the US Food and Drug Administration [FDA]), which employs a superimposed alternating and direct electric current, named as a Proxywave^®^ technology, similar to the intensity of the biocurrent, in plaque removal and reducing gingivitis by biofilm removal through the bioelectric effect. This study enrolled 40 volunteers with gingivitis. Dental observations were made every two weeks, before and after the use of each toothbrush. We randomly assigned participants into two groups: one group used the Proxywave^®^ toothbrush (PB) for two weeks followed by the control toothbrush (CB) for two weeks, while the other group used the CB for two weeks followed by the PB. The participants had a two-week washout period. If the toothbrush used earlier has had an effect on the bacterial flora in the oral cavity, this is to remove this effect and return it to its previous state. During each dental visit, we recorded plaque index (PI) and gingival index (GI) scores. The PI and GI scores were significantly lower in both the PB and the CB (*p* < 0.05). Considering the PI, there was no significant difference between the toothbrushes on all the surfaces. Considering the GI, the PB showed a significant decrease in the interproximal surface, compared to the CB (*p* < 0.05). The PB showed a significant decrease in the interproximal GI and had a beneficial effect in the interproximal area where the bristles could not reach. No adverse events were observed in the participants during the clinical trial. The microcurrent toothbrush is a device that can be safely used for plaque removal.

## 1. Introduction

Plaque is defined as a deposit, mainly composed of bacteria, that forms a biofilm by adhering to the tooth surfaces, restorations, and other prosthetic appliances [1]. Experimental gingivitis studies by Löe et al. identified that plaque is the root cause of the progression of gingivitis and periodontitis, which presents as the inflammation and destruction of periodontal tissues. Accumulation of bacterial plaque causes clinical signs of gingivitis within 10 days [2,3]. Hence, to control and prevent periodontal disease, it is necessary to prevent plaque formation on the tooth surface or eliminate it before inflammatory changes occur in the gingiva [4]. Effective daily bacterial plaque removal is essential to control periodontal disease [5].

The most common and reliable method of plaque control is mechanically cleaning the oral tissues using a toothbrush and oral hygiene aids [6,7]. Since its public introduction in the 1960s, the electric toothbrush has greatly advanced in its shape and movement pattern. However, despite improvements in toothbrush types and designs, most people are only able to remove ~50% of plaque while brushing [8]. This warrants the development of a toothbrush that can remove more plaque daily.

The Proxywave^®^ toothbrush (proxyhealthcare, Ulsan, Korea) used in this study integrates a new electromagnetic wave for effective biofilm removal. The wave, termed Proxywave^®^, features a combined mechanism of alternating and direct currents (AC and DC) for biofilm reduction. Biofilms contain polysaccharides and bacterial cells that are electrically polarized [9,10,11]. Thus, the external electric stimulation may disturb the structure of the biofilm and interrupt the bacterial metabolic status [12,13]. The AC, in particular, increases the porosity of the biofilm structure due to specific frequency vibration induction [14], and the DC changes the electrolyte condition, which is critical to maintaining cell metabolism [13]. For instance, the AC can facilitate the delivery of antibiotics into the biofilm and the DC field can change the local pH, resulting in increased cell detachment. The “bioelectric effect” results when combined electrical-based treatments increase the efficacy of antibiotics on biofilms [15,16]. The Proxywave^®^ combines both AC and DC currents, and this synergistic mechanism significantly improves biofilm treatment efficacy [17,18].

The purpose of this study was to compare the effects of a toothbrush applying the Proxywave^®^ technology (Proxywave^®^ toothbrush, PB), with that of a control toothbrush (CB) in plaque removal and reducing gingival inflammation, and to evaluate its efficacy on the difficult-to-reach interproximal and lingual surfaces.

## 2. Material and Methods

### 2.1. Study Design

This study was an examiner-blind, randomized study with a cross-over design. Each participant received one test brush, one control brush, and toothpaste (Tromatz toothpaste, Sungwon Co., Ltd., Goyang, Korea). We randomly divided the participants into two groups and examined them four times at two-week intervals (Figure 1). It was impossible to examine all 40 participants on the same day. Therefore, in order to examine at different time points, 40 people were randomly divided into group 1 (*n* = 20) and group 2 (*n* = 20). That is, group 1 and group 2 are not classifications for data analysis. As a crossover study in which one person used both the test brush and the control brush, the data were analyzed by dividing into pre- and post-brushing of the test brush (*n* = 40) and pre- and post-brushing of the control brush(*n* = 40). In order to blind the examiner, randomization, enrollment of participants, and toothbrush distribution were made and maintained by the research assistant. Group 1 used the Proxywave^®^ toothbrush (PB) for two weeks followed by the control toothbrush (CB) for two weeks, while Group 2 used the CB for two weeks followed by the PB.

Since plaque control ability varies greatly among individuals, this study was designed as a cross-over design that allows one person to use both the test brush and the control brush without dividing it into a control group and a test group. Thus, to visualize the degree of plaque removal by the toothbrush, we instructed participants not to alter their frequency, duration, nor method of toothbrushing. During the two-week washout period, the participants used their previous toothbrush to exclude the effect of brushing with the PB. During the study period, we prohibited the use of oral hygiene products other than the provided toothbrush.

### 2.2. Brush

Using the PB, this study evaluated the Proxywave^®^, which consists of 10 MHz of AC and 0.5 V DC. Proxywave^®^ induces a microcurrent of 100 µA, a signal similar to a biocurrent, which is safe for human applications [19]. The brush is comprised of two parallel electrodes that introduce the wave directly to the teeth. The wave is effective in biofilm reduction at a ~10 mm distance from the electrode position [18]. Both the control brush (CB, Baekjae Industry, Icheon, Korea) and the test brush (Tromatz basic^®^, proxyhealthcare, Ulsan, Korea) are soft and multi-tufted. The bristles are 12 mm long and 0.15 mm in diameter. It is made from Polybutylene terephthalate. The device overview is shown in Figure 2. Aside from the two electrodes on the brush, the overall shape and usage are identical to a traditional toothbrush.

### 2.3. Participants

For this study, we selected 40 volunteers (non-dental students) who provided informed consent. The recruitment through advertisements was initiated in September 2020 and all the study visits were completed by December 2020. The study was approved by the Institutional Review Board (IRB) of Ulsan University Hospital (IRB protocol No: 2020-09-020-003). Medically healthy subjects between 19 and 65 years of age were recruited. Inclusion criteria included a minimum of 20 natural teeth, mild to moderate gingivitis, no calculus deposition and staining. Exclusion criteria were individuals who were pregnant or lactating, smokers, a history of usage of antibiotics during the 3 months prior to the study and presence of acute oral lesions, orthodontic appliance, partial denture, malocclusion.

### 2.4. Clinical Measurements

At each visit, we measured the Gingival Index (GI) of Löe and Silness and the Turesky modification of the Quigley–Hein plaque index (PI) for all teeth [19,20,21]. We measured the GI using UNC 15 probes across four tooth surfaces (buccal, lingual, mesial, and distal). We applied a disclosing agent (RED-COTE^®^, Sunstar Americas, Inc., Chicago, IL, USA) to all the teeth to visualize the plaque present on the tooth surfaces. After washing with water for approximately 15 s, we measured the PI at six sites per tooth (mesiobuccal, buccal, distobuccal, mesiolingual, lingual, and distolingual). The same researcher performed the evaluation and measurement of the PI and GI to reduce inter-researcher error.

### 2.5. Data Analysis

We averaged the PI and GI by dividing the average score of all teeth for each brush type, interproximal and lingual, respectively. Additionally, since we conducted the study in a crossover design, we divided the analysis into treatment effects and periodic effects, and considered *p*-values ≤ 0.05 statistically significant. We performed the paired *t*-test and chi-square test using the SPSS statistics version 24 software (IBM, Armonk, NY, USA).

### 2.6. Safety Assessment

At each visit, we assessed the safety of the toothbrushing protocol by examining the hard and soft tissues. The hard tissue was checked for influence on the dental restoration and hypersensitivity to the cervical area. Soft tissue was examined for signs of oral pathology and irritation to the hard and soft palate, gingivae, buccal mucosa, sublingual space areas, and tongue. All changes observed at the oral examination were recorded in the case report form.

Microcurrents below 100 µA are harmless to the human body [22,23]. PB has been approved by the FDA, US Federal Communications Commission, Korea Certification mark (Korea), Conformite European mark (Europe), Product Safety Electrical mark (Japan) as safe.

## 3. Results

Forty participants completed the trial and no dropouts were observed. The mean age of the participants was 37.58 ± 7.43, with 12 males and 28 females. The mean brushing duration was 2.46 ± 0.71 min, and the brushing frequency varied from 2−5 times a day (mean, 3.15 ± 0.58) (Table 1).

We measured PI and GI indices on each tooth surface across the entire dentition, and calculated values on the interproximal and lingual surfaces to evaluate the effect on the hard-to-reach areas (Table 2 and Table 3). Table 2 shows the pre- and post-brushing PI and GI scores of the PB and CB. Considering the treatment effect of PB and CB, both toothbrushes showed a significant decrease in PI and GI after brushing (*p* < 0.05) (Table 2). Table 3 shows the reduction in the PI and GI as the difference before and after each toothbrush use. There was no significant difference between the PI of the CB and the PB, and this was true of all tooth surfaces, including the interproximal and lingual surfaces (*p* > 0.05). Regarding the GI, the PB showed a significant decrease in the interproximal surface GI compared to the CB (*p* < 0.05) (Table 3). There was no statistically significant difference in terms of a residual effect. (*p* > 0.05).

### Safety

Participant reports and oral examinations indicated no serious adverse events related to toothbrush use. No participant reported difficulty in using his or her assigned brush. We occasionally observed gingival abrasion, most likely resulting from traumatic toothbrushing. Therefore, the PB could be used safely like a normal CB for optimal oral hygiene care.

## 4. Discussion

There was no difference in the PI between the CB and PB; however, we observed a difference in the GI. Reportedly, the onset of gingival inflammation is influenced not only by the amount of plaque, but also by other factors. A previous study observed fibroblast proliferation only in areas with mechanical irritation due to brushing [19]. The therapeutic effect of microcurrent has proven to reduce inflammation, tissue regeneration such as bone regeneration and wound healing, and increase growth factor expression [24,25]. Our findings also suggest that gingival cell proliferation was activated by the Proxywave^®^ as well as mechanical stimulation by brushing.

The hypothesized mechanism of action behind the bioelectrical effect is based on an external electric field altering a bacterial cell membrane containing various partially charged molecules, including cell proteins, polysaccharides, nucleic acids, and lipids [26,27]. Proxywave^®^ is a bioelectromagnetic wave specialized for biofilm removal, developed while studying the relationship between bioelectric effects and electric signals. While the existing method was focused on a single mechanism of DC or AC, Proxywave^®^ maximizes the effect by combining the two signals. When DC and AC are simultaneously applied to the biofilm, the metabolic stress in the biofilm increases rapidly, inducing the bioelectric effect based on the electrostatic force, media electrolysis, inactivation of enzyme, non-uniform electrolyte distribution and electrochemical environmental changes, resulting in a reduction of biofilm [17,18].

The bioelectric effect is recognized as a promising method for enhancing the efficacy of antibiotics in biofilms by combining electrical application with low-dose antibiotic treatment [12]. In fact, the bacterial resistance to antibiotics increases by about 1000 times among bacteria in a biofilm state than bacteria in a suspended state [28]. Future studies should evaluate the effectiveness of a subantimicrobial dose of doxycycline (20 mg doxycycline twice daily) in combination with a Proxywave^®^ toothbrush as an adjunctive treatment in patients with periodontitis [29].

Studies comparing control and electric toothbrushes have mainly been conducted to evaluate differences in plaque and gingivitis reduction, or to establish the superiority of the electric toothbrush over CB. Although several studies have deemed electric toothbrushes superior, they cannot be applied universally [30].

To compensate for these contradictory experimental results, we performed a paired *t*-test comparison. We did not restrict individual characteristics such as brushing ability, since we did not divide the subjects into a control group and a test group. Instead, we calculated the PI and GI for each individual.

We conducted this study using a crossover design, which can be implemented with only half of the participants compared to other study designs; however, it possesses the possibility of residual effects. To reduce the residual effect, we implemented a two-week washout period in the study. The result of the statistical analysis was not significant.

According to Kleber et al., about 40% of the entire tooth surface is not properly regulated, the posterior teeth are more plaque prone than the anterior teeth, and the lingual and proximal surfaces are more plaque prone than the buccal surfaces [31,32,33]. Generally, these areas are highly associated with gingivitis and periodontitis. Therefore, it is necessary to study a toothbrush designed for self-care and plaque control on the lingual and proximal surfaces, which are difficult areas to reach [34].

In this study, we only tested supragingival plaque, and did not evaluate subgingival plaque. This study only targeted patients with gingivitis. Future studies should evaluate whether long-term use of the PB can regenerate periodontal tissue in patients with periodontitis. Moreover, there is a need for additional research on the size of the toothbrush head and bristles, considering the length of time in which the Proxywave^®^ could maximize the effect in the area receiving electromagnetic waves.

## 5. Conclusions

Both CB and PB showed a significant reduction in plaque and gingival inflammation following brushing, and the interproximal GI was significantly decreased in the PB. In this study, we found that the PB may improve inflammation in the interdental area, compared to the CB. Therefore, the PB may be safely used for plaque removal and reduction of gingivitis.

## Figures and Tables

**Figure 1 ijerph-18-08255-f001:**
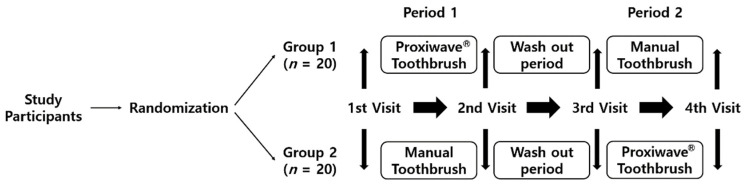
Flow chart of the study.

**Figure 2 ijerph-18-08255-f002:**
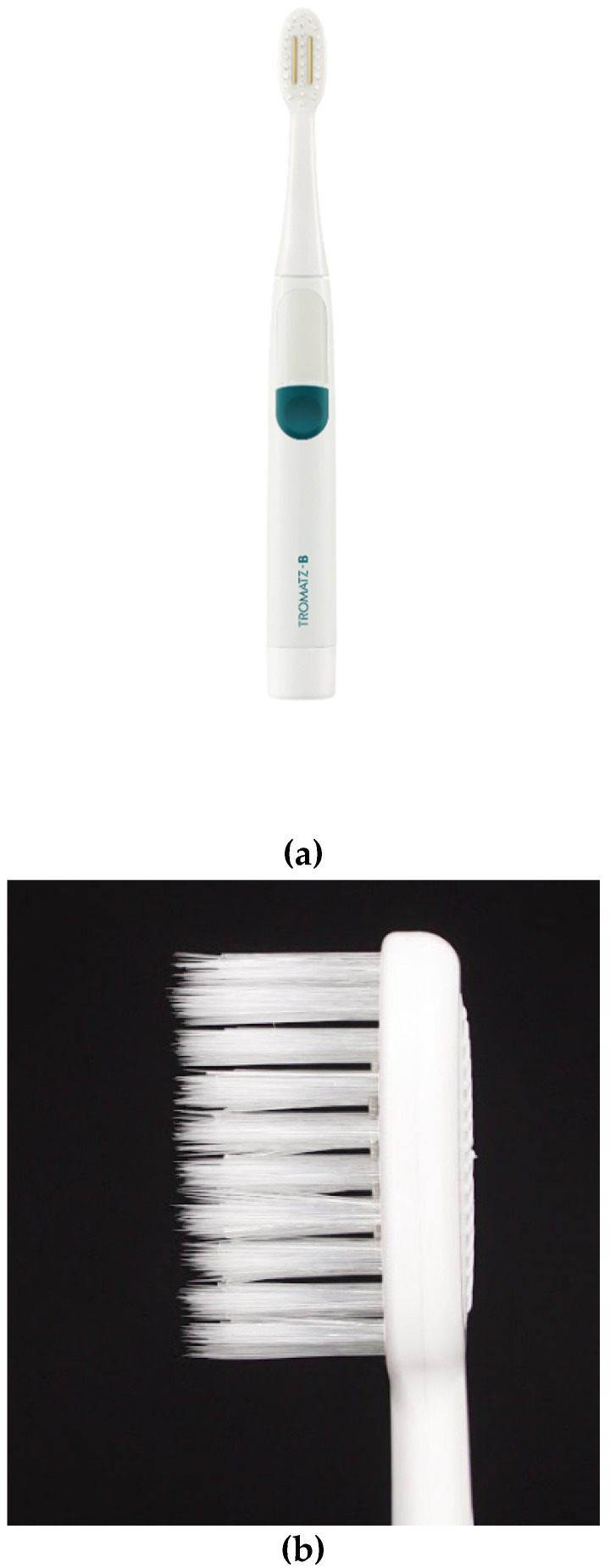
(**a**) Proxywave^®^ test brush, (**b**) Side view of the Proxywave^®^ toothbrush, (**c**) Side view of the control toothbrush (Baekje Industry) brush is identical to the tested toothbrush.

**Table 1 ijerph-18-08255-t001:** Demographic of participants.

	Total (*n* = 40)	Group 1 (*n* = 20)	Group 2 (*n* = 20)	*p*-Value
Age	37.58 ± 7.43	37.70 ± 7.15	37.45 ± 7.88	0.917
Sex				1.000
Male	12 (30.0)	6 (30.0)	6 (30.0)	
Female	28 (70.0)	14 (70.0)	14 (70.0)	
Brushing time (minutes)	2.46 ± 0.71	2.60 ± 0.60	2.33 ± 0.80	0.226
Brushing frequency	3.15 ± 0.58	3.15 ± 0.67	3.15 ± 0.49	1.000

Continuous variables were compared using the independent *t*-test, and categorical variables were compared using the Chi-square test.

**Table 2 ijerph-18-08255-t002:** Comparison of plaque index and gingival index before and after brushing: mean score (standard deviation) for all, interproximal, and lingual surfaces.

	Plaque Index (PI)			Gingival Index (GI)		
	Pre-Brushing	Post-Brushing	*p*-Value	Pre-Brushing	Post-Brushing	*p*-Value
All surface						
PB	1.24 ± 0.36	0.88 ± 0.27	0.000	0.22 ± 0.19	0.11 ± 0.11	0.000
CB	1.22 ± 0.33	0.93 ± 0.33	0.000	0.19 ± 0.16	0.12 ± 0.13	0.000
Interproximal surface						
PB	1.41 ± 0.36	1.05 ± 0.28	0.000	0.28 ± 0.24	0.13 ± 0.16	0.000
CB	1.40 ± 0.34	1.11 ± 0.36	0.000	0.23 ± 0.21	0.15 ± 0.17	0.000
Lingual surface						
PB	1.36 ± 0.42	1.02 ± 0.36	0.000	0.22 ± 0.19	0.13 ± 0.12	0.001
CB	1.36 ± 0.41	1.05 ± 0.43	0.000	0.22 ± 0.17	0.15 ± 0.16	0.000

PB, Proxywave^®^ toothbrush (*n* = 40); CB, Control toothbrush (*n* = 40); Statistical analysis method: Paired *t*-test.

**Table 3 ijerph-18-08255-t003:** Comparison of difference between plaque index (PI) and gingival index (GI) before and after brushing by toothbrush type: mean score (standard deviation) for all, interproximal, and lingual surfaces.

	Reduction Plaque Index (PI)			Reduction Gingival Index (GI)		
	PB	CB	*p*-Value	PB	CB	*p*-Value
All surface	−0.36 ± 0.29	−0.28 ± 0.29	0.252	−0.11 ± 0.12	−0.07 ± 0.09	0.062
Interproximal surface	−0.36 ± 0.31	−0.29 ± 0.32	0.307	−0.14 ± 0.16	−0.08 ± 0.13	0.034
Lingual surface	−0.33 ± 0.31	−0.31 ± 0.36	0.75	−0.08 ± 0.14	−0.07 ± 0.12	0.673

PB, Proxywave^®^ toothbrush (*n* = 40); CB, Control toothbrush (*n* = 40); Statistical analysis method: Paired *t*-test.

## Data Availability

Original data are available upon request to the corresponding author.

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
