# Peer review of "Bioelectric Effect of a Microcurrent Toothbrush on Plaque Removal"

_ijerph, 2021, doi:10.3390/ijerph18168255_

Round 1

Reviewer 1 Report

The study evaluates the efficacy of microcurrent toothbrush vs a manual toothbrush.

Overall, the study is clearly presented but I have some major concerns. Please see comments below.

Title: As mentioned in the title, I don’t see any bioelectric effect tested in the study. Instead it is a study trying to identify which toothbrush is the better one for reducing plaque and gingivitis. If you would have wanted to test the efficacy of the bioelectric effect, the best control should have been the PB, but without the bioelectric effect. Otherwise, change the title.

Line 11: Why do you have to mention Proxywave technology here. Is it just a  name used for marketing right? Instead talk about the bioelectric technology, what it is and about its efficacy in plaque removal.

Line 16: What is meant by wash out period. Explain more clearly

Line 16. Explain at what time points in the study the dental visits/observations were made. Need to be clear.

Line 22: FDA approved means its already recommended for use right?. You don’t have to support/recommend it is safe for use. Anyway there is no data available that supports its safety.

Line 27: This statement is not true. You cannot find plaque in gingiva or other tissues. I don’t see anything mentioned like that in the cited paper either. Please correct the statement.

Line 47: Remove #12 citation. Doesn’t seem to support the statement.

Line 50: You need to back this statement with proper references. The references included does not support the statement.

Line 68: Is 2 weeks enough to develop plaque or gingivitis? How do you know the level of plaque or gingivitis which occurred during the previous PB and MB group would not affect the PI or GI in the second MB and PB groups, respectively.

Fig 1: Indicate exactly, at which point the dental visits/examinations were done. Currently it is not clear.

Line 80 and Line 121: Here and elsewhere, use proper symbols. Shouldn’t it be µA ?

Line 87: I think you should have used the same brush without the electrodes. Otherwise, you are not comparing the efficacy of the proxywave/bioelectric effect but only the tooth brushes.

Line 165: Provide examples for molecules and proteins...

Line 170: What is plummetic effect? Explain

Line 176: What is so special about doxycycline that it “should” be evaluated? Is it just an example for the use of antibiotics as a combination therapy with proxywave? If so, provide it just as an example.

Line 184: I don’t see this data. I can only see the mean values for the groups in Table 2&3. Please explain more clearly how you calculated these numbers.

Line 189: Statistical analysis of what?

Author Response

Response to Reviewer 1 Comments

Title: As mentioned in the title, I don’t see any bioelectric effect tested in the study. Instead it is a study trying to identify which toothbrush is the better one for reducing plaque and gingivitis. If you would have wanted to test the efficacy of the bioelectric effect, the best control should have been the PB, but without the bioelectric effect. Otherwise, change the title.

Thank you for your valuable suggestion. The toothbrush used in this study is a toothbrush with ProxywaveⓇ technology similar to biocurrent, and it is a method of removing biofilm with the bioelectric effect.

Line 11: Why do you have to mention Proxywave technology here. Is it just a  name used for marketing right? Instead talk about the bioelectric technology, what it is and about its efficacy in plaque removal.

We revised the manuscript to define the specialized signal as the Proxywave and then cite it at the rest of manuscript.

“This study evaluated the effectiveness of a microcurrent toothbrush (approved by the US Food and Drug Administration [FDA]), which employs a superimposed alternating and direct electric current, named as a ProxywaveⓇ technology, similar to the intensity of the biocurrent, in plaque removal and reducing gingivitis by biofilm removal through the bioelectric effect.”

“The Proxywave toothbrush used in this study integrates a new electromagnetic wave for effective biofilm removal. The wave, termed Proxywave, features a combined mechanism of alternating and direct currents (AC and DC) for biofilm reduction.”

Line 16: What is meant by wash out period. Explain more clearly

The wash out period is 10 to 14 days. During this period, participants brush their teeth with their original toothbrush. The reason is to rule out the influence of the toothbrush you used before.

 If the toothbrush used earlier has had an effect on the bacterial flora in the oral cavity, this is to remove this effect and return it to its previous state.

Line 16. Explain at what time points in the study the dental visits/observations were made. Need to be clear.

The recruitment through advertisements was initiated in September 2020 and all the study visits were completed by December 2020. We revised the manuscript. (2.3 Participants)

Line 22: FDA approved means its already recommended for use right?. You don’t have to support/recommend it is safe for use. Anyway there is no data available that supports its safety.

In the manuscript, we have mentioned about the intensity of the microcurrent, which is approximately 100μA. The intensity has been utilized on medical treatment specially treatment of inflammation. We added a reference article which states the safety of electric stimulation below 1000μA.

[19] Lee, J., Yoon S., Kim, T., Park S. The effects of microcurrents on inflammatory reaction induced by ultraviolet irradiation, J. Phys. Ther. Sci. 2011, 23: 693-696.

Line 27: This statement is not true. You cannot find plaque in gingiva or other tissues. I don’t see anything mentioned like that in the cited paper either. Please correct the statement.

Dental plaque can form on any surface in the oral cavity, even with subgingiva. But we revised more proper statement.

Line 47: Remove #12 citation. Doesn’t seem to support the statement.

 The reference paper has been revised to the relevant article.

[12] Stoodley, P.; Sauer, K.; Davies, D.G.; Costerton, J.W. Biofilms as complex differentiated communities. Annu Rev Microbiol 2002,56,187–209.

à [12] Del Pozo, J.; Rouse, M.; Patel R. Bioelectric effect and bacterial biofilms. A systematic review. Int. J. Artif. Organs. 2008, 31(9), 786 -95

Line 50: You need to back this statement with proper references. The references included does not support the statement.

We have revised with proper reference.

“The AC, in particular, increases the porosity of the biofilm structure due to specific frequency vibration induction [14], and the DC changes the electrolyte condition, which is critical to maintaining cell metabolism [14] [13].”

 Line 68: Is 2 weeks enough to develop plaque or gingivitis? How do you know the level of plaque or gingivitis which occurred during the previous PB and MB group would not affect the PI or GI in the second MB and PB groups, respectively.

Experimental gingivitis studies by Löe et al. identified that plaque is the root cause of the progression of gingivitis and periodontitis, which presents as the inflammation and destruction of periodontal tissues. Accumulation of bacterial plaque causes clinical signs of gingivitis within 10 days [2,3]

Washout period was held to eliminate the effects of previous toothbrush.

Since plaque control ability varies greatly among individuals, this study was designed as a cross over design that allows one person to use both the test brush and the control brush without dividing it into a control group and a test group. Thus, to visualize the degree of plaque removal by the toothbrush, we instructed participants not to alter their frequency, duration, nor method of toothbrushing.

We did not restrict individual characteristics such as brushing ability, since we did not calculate the average value for each group. Instead, we calculated the PI for each individual and compared them separately.

 Fig 1: Indicate exactly, at which point the dental visits/examinations were done. Currently it is not clear.

At each visit, we measured the Gingival Index (GI) of Löe & Silness and the Turesky modification of the Quigley–Hein plaque index (PI) for all teeth [19–21]. 

Line 80 and Line 121: Here and elsewhere, use proper symbols. Shouldn’t it be µA ?

We revised it.

Line 87: I think you should have used the same brush without the electrodes. Otherwise, you are not comparing the efficacy of the proxywave/bioelectric effect but only the tooth brushes.

We attached the control brush figure. The control brush has the same bristles as the test brush.

Line 165: Provide examples for molecules and proteins...

 Revised the manuscript.

“The hypothesized mechanism of action behind the bioelectrical effect is based on an external electric field altering a bacterial cell membrane containing various partially charged molecules, including cell proteins such as polysaccharides, nucleic acids, proteins, lipids, and lipoproteins [27,28].”

Line 170: What is plummetic effect? Explain

Revised the manuscript with more details of explanation.

While the existing method was focused on a single mechanism of DC or AC, Proxywave maximizes the effect by combining the two signals. When DC and AC are simultaneously applied to the biofilm, the metabolic stress in the biofilm increases rapidly, inducing a plummeting effect due to the electrostatic force induced by the external superimposed AC and DC electric field [17,18].

Line 176: What is so special about doxycycline that it “should” be evaluated? Is it just an example for the use of antibiotics as a combination therapy with proxywave? If so, provide it just as an example.

The bioelectric effect is recognized as a promising method for enhancing the efficacy of antibiotics in biofilms by combining electrical application with low-dose antibiotic treatment. Subantimicrobial dose doxycycline (SDD--20 mg doxycycline twice daily) is indicated as an adjunctive treatment for periodontitis. 

Preshaw PM, Hefti AF, Jepsen S, Etienne D, Walker C, Bradshaw MH. Subantimicrobial dose doxycycline as adjunctive treatment for periodontitis. A review. J Clin Periodontol. 2004

Line 184: I don’t see this data. I can only see the mean values for the groups in Table 2&3. Please explain more clearly how you calculated these numbers.

 I was impossible to examine all 40 participants on the same day and this study was examiner-blind. Therefore, in order to examine at different time points, 20 people were randomly divided into group 1 and group 2, and there is no clinical significance. As a crossover study in which one person used both the test brush and the control brush, the data were analyzed by dividing into pre- and post-brushing of the test brush(n=40) and pre- and post-brushing of the control brush(n=40). We revised the manuscript (2.3 Study design & Table 2, 3 )

Line 189: Statistical analysis of what?

We performed the paired t-test and Chi-square test using the SPSS statistics version 24 software (IBM, Armonk, NY, USA).

Power                N        Diff0         Diff1        Alpha         Beta   SdPeriod         Size

0.8134                  40         0.120          0.490            0.05        0.1866            0.40          0.654

Summary Statements

A two-sided t-test achieves 81% power to infer that the mean difference is not 0.120 when the total sample size of a 2x2 cross-over design is 40, the actual mean difference is 0.490, the standard deviation of the period differences for each subject within each sequence is 0.400, and the significance level is 0.05.

References

Chow, S.C. and Liu, J.P. 1999. Design and Analysis of Bioavailability and Bioequivalence Studies. Marcel Dekker. New York

Chow, S.C.; Shao, J.; Wang, H. 2003. Sample Size Calculations in Clinical Research. Marcel Dekker. New York.

Julious, Steven A. 2004. 'Tutorial in Biostatistics. Sample sizes for clinical trials with Normal data.' Statistics in Medicine, 23:1921-1986.

Senn, Stephen. 2002. Cross-over Trials in Clinical Research. Second Edition. John Wiley & Sons. New York.

Sample size calculator program

Hintze, J. (2011). PASS 11. NCSS, LLC. Kaysville, Utah, USA. www.ncss.com.

Reviewer 2 Report

The authors have reported a microcurrent tooth brush in biofilm removal leading to the successful reduction of plaque. The authors contributed a significant amount to the experiments and data analysis and are highly commendable. The findings of this article seem to be interesting and I recommend the manuscript for publication with minor changes.

  1. In figure 2 along with the Proxywave tooth brush picture, keep a zoom-in the picture of brush bristles alone as figure 2B. Also if possible keep a picture of a manual toothbrush.
  2. It is confusing to which group (1 or 2) the pre and post-measurement were done? Make it clear for the readers to understand.

Author Response

Response to Reviewer 2 Comments

  1. In figure 2 along with the Proxywave tooth brush picture, keep a zoom-in the picture of brush bristles alone as figure 2B. Also if possible keep a picture of a manual toothbrush.
  2. It is confusing to which group (1 or 2) the pre and post-measurement were done? Make it clear for the readers to understand.

Thank you for your valuable suggestion

  1. We attached the control brush figure. The control brush has the same bristles as the test brush.

  1. I was impossible to examine all 40 participants on the same day and this study. Therefore, in order to examine at different time points, 20 people were randomly divided into group 1 and group 2, and there is no clinical significance. As a crossover study in which one person used both the test brush and the control brush, the data were analyzed by dividing into pre- and post-brushing of the test brush(n=40) and pre- and post-brushing of the control brush(n=40).

We revised the manuscript (2.3 Study design & Table 2, 3)

Reviewer 3 Report

This clinical trial tested the plaque removal effect of a micro-current toothbrush by comparing it with a normal toothbrush. The study design is good and the topic is interesting. Here are my comments on this manuscript.

Since the brush shape can be an important confounding factor, the test and control toothbrush should have a similar brush shape. In addition to the picture of the micro-current toothbrush, please also include the photo of the control toothbrush. A front photo and a side photo should be provided for each toothbrush.

I suggest changing the word "manual toothbrush" into "control toothbrush". Actually, both these two kinds of toothbrushes are manual toothbrushes. Calling one of them a "manual toothbrush" is confusing. This may make readers think the other toothbrush is a vibrate/rotary electric toothbrush.

Please add descriptions about when the visit was performed in the Material and Method section.

While the study design is good, the presentation of the results is confusing.

In the Material and Method section, the two groups are named group 1 and group 2. However, in the Results, they are called Proxywave Toothbrush group (PB) and Manual Toothbrush Group (MB). This is not consistent. Actually, these two kinds of toothbrushes, PB and MB, were both used in group 1 and group 2. This makes it non-sense to call the two groups PB or MB. Please use another way to name the groups and organize the results.

There were four PI/GI measuring time points. However, the results only reported two time points (before and after). This confusing. Please revise.

Author Response

Thank you for your valuable suggestion.

We revised the manual toothbrush to a control toothbrush.

I was impossible to examine all 40 participants on the same day. Therefore, in order to examine at different time points, 40 people were randomly divided into group 1 and group 2, and there is no clinical significance. As a crossover study in which one person used both the test brush and the control brush, the data were analyzed by dividing into pre- and post-brushing of the test brush(n=40) and pre- and post-brushing of the control brush(n=40). We revised the manuscript (2.3 Study design & Table 2, 3)

Round 2

Reviewer 1 Report

Although many of the issues are addressed, there are few more suggestions that I believe might improve the manuscript and would benefit the readership.

Please see comments below.

Although the pictures of the toothbrushes look similar, maybe you can provide more similar characteristics between the two toothbrushes that supports the argument. Ex: type/make of bristles, dimensions/length?

Line 16: Explaining (briefly) the term ‘wash out period’ might be helpful for readers who are not familiar with clinical trials.

Also include this sentence from your answer somewhere appropriate in the manuscript (maybe method or discussion), “If the toothbrush used earlier has had an effect on the bacterial flora in the oral cavity, this is to remove this effect and return it to its previous state”.

Line 16. Explain at what time points in the study the dental visits/observations were made. “Dental observations were made every two weeks, before and after the use of each tooth brush…….”, something in that line. Including a sentence like this would add more clarity to the methodology described in the abstract.

Line 22: Include a summary sentence stating some of the data related to safety that you observed. This will support the “safety” aspect of the toothbrush.

Fig 1: Indicate exactly, at which point the dental visits/examinations were done. Simply, draw two arrows arrow between 1st visit and a point infront of ‘Proxywave toothbrush’ and ‘manual toothbrush’ boxes. Also draw an arrow between 2nd visit and the point in between “Proxywave toothbrush’ and “washout period” boxes, and similarly another arrow down, to a point in between “manual toothbrush’ and “washout period” boxes. Do the same for the 3rd and 4th visit.

Line 89 There should be no dash (-) in 10 mm. Please correct units, symbols and typos throughout the manuscript.

Line 172: “The hypothesized mechanism of action behind the bioelectrical effect is based on an external electric field altering a bacterial cell membrane containing various partially charged molecules, including cell proteins such as polysaccharides, nucleic acids, proteins, lipids, and lipoproteins [27,28].”

Remove “such as” , otherwise it looks as if polysaccharides, nucleic acids, proteins, lipids, and lipoproteins are proteins, which is inaccurate. Also please correct any grammar throughout the manuscript.

Line 180: What is plummetic effect? Still the sentence doesn’t explain what it is.

Line 182: About the use of doxycycline, please include the reference in the manuscript, the one you have mentioned in your answer.

Line 184: I was impossible to examine all 40 participants on the same day and this study was examiner-blind. Therefore, in order to examine at different time points, 20 people were randomly divided into group 1 and group 2, and there is no clinical significance.

 “………and there is no clinical significance”, not sure what you mean here. Please correct the grammar.

Line 200: Statistical analysis of what? I meant it is not clear between which two samples that you are comparing. Please indicate this.

Author Response

Thanks to your valuable suggestions, I was able to help the reader understand my article and fill in the gaps. Thank you very much.

Although the pictures of the toothbrushes look similar, maybe you can provide more similar characteristics between the two toothbrushes that supports the argument. Ex: type/make of bristles, dimensions/length?

Line 95  Both the control brush (CB, Baekjae Industry, Icheon, Korea) and the test brush (Tromatz basic, proxyhealthcare, Ulsan, Korea) are soft and multi tufted.

We have added details of the information.

The specification of the brush is summarized in below.

Diameter of bristles = 0.15mm

Length of bristles = 12mm

Material: Polybutylene terephthalate (PBT)

Line 16: Explaining (briefly) the term ‘wash out period’ might be helpful for readers who are not familiar with clinical trials.

Also include this sentence from your answer somewhere appropriate in the manuscript (maybe method or discussion), “If the toothbrush used earlier has had an effect on the bacterial flora in the oral cavity, this is to remove this effect and return it to its previous state”.

We have included that sentence in the abstract.

Line 16. Explain at what time points in the study the dental visits/observations were made. “Dental observations were made every two weeks, before and after the use of each tooth brush…….”, something in that line. Including a sentence like this would add more clarity to the methodology described in the abstract.

We have included that sentence in the abstract.

Line 22: Include a summary sentence stating some of the data related to safety that you observed. This will support the “safety” aspect of the toothbrush.

We have included the sentence about “safety” in the abstract.

Fig 1: Indicate exactly, at which point the dental visits/examinations were done. Simply, draw two arrows arrow between 1st visit and a point infront of ‘Proxywave toothbrush’ and ‘manual toothbrush’ boxes. Also draw an arrow between 2nd visit and the point in between “Proxywave toothbrush’ and “washout period” boxes, and similarly another arrow down, to a point in between “manual toothbrush’ and “washout period” boxes. Do the same for the 3rd and 4th visit.

 We revised the Fig 1.

Line 89 There should be no dash (-) in 10 mm. Please correct units, symbols and typos throughout the manuscript.

 We revised it.

Line 172: “The hypothesized mechanism of action behind the bioelectrical effect is based on an external electric field altering a bacterial cell membrane containing various partially charged molecules, including cell proteins such as polysaccharides, nucleic acids, proteins, lipids, and lipoproteins [27,28].”

Remove “such as” , otherwise it looks as if polysaccharides, nucleic acids, proteins, lipids, and lipoproteins are proteins, which is inaccurate. Also please correct any grammar throughout the manuscript.

 The manuscript has been revised accordingly.

The hypothesized mechanism of action behind the bioelectrical effect is based on an external electric field altering a bacterial cell membrane containing various partially charged molecules, including cell proteins, polysaccharides, nucleic acids, and lipids [27,28]

Line 180: What is plummetic effect? Still the sentence doesn’t explain what it is.

 The manuscript has been revised. We have changed the word from “the plummeting effect” to “bioelectric effect” which is defined as the electric field induced biofilm reduction mechanism.

“When DC and AC are simultaneously applied to the biofilm, the metabolic stress in the biofilm increases rapidly, inducing the bioelectric effect based on the electrostatic force, media electrolysis, inactivation of enzyme, non-uniform electrolyte distribution and electrochemical environmental changes, resulting in reduction of biofilm [17,18].”

Line 182: About the use of doxycycline, please include the reference in the manuscript, the one you have mentioned in your answer.

We added the reference to the manuscript.

Line 184: I was impossible to examine all 40 participants on the same day and this study was examiner-blind. Therefore, in order to examine at different time points, 20 people were randomly divided into group 1 and group 2, and there is no clinical significance.

 “………and there is no clinical significance”, not sure what you mean here. Please correct the grammar.

We revised the sentence. Line 69

Line 200: Statistical analysis of what? I meant it is not clear between which two samples that you are comparing. Please indicate this.

We performed the paired t-test and Chi-square test using the SPSS statistics version 24 software (IBM, Armonk, NY, USA). A paired t test was used to compare the plaque removal effects of the testbrush and control brush. Since test brush and control brush are different measurement tools, they were regarded as independent groups. The comparison values ​​before and after treatment were analyzed as paired data.